# Reporting Liver Cancer Trends in the Island of Crete, Greece: Results from a Geo-Epidemiological Study

**DOI:** 10.3390/ijerph191610166

**Published:** 2022-08-17

**Authors:** Stavros Kalpadakis, Dimitra Sifaki-Pistolla, Emmanouil K. Symvoulakis, Panagiotis Kelefiotis-Stratidakis, Lambros Vamvakas, Dimitrios Mavroudis, Christos Lionis

**Affiliations:** 1Clinic of Social and Family Medicine, Faculty of Medicine, University of Crete, 71003 Heraklion, Greece; 2Department of Chemistry, Voutes University Campus, University of Crete, 70013 Heraklion, Greece; 3Pathology Oncology Clinic, University Hospital of Heraklion, 71003 Heraklion, Greece; 4School of Medicine, University of Crete, 71003 Heraklion, Greece

**Keywords:** liver cancer, incidence, mortality, cancer epidemiology, Geographical Information Systems

## Abstract

Liver cancer is one of the most frequent cancers in Europe and Greece. An increase in specific risk factors, such as metabolic syndrome and obesity, has been observed in Greece. Therefore, exploring temporal trends of liver cancer incidence and mortality is crucial. This study aims to assess the “burden” of malignant liver tumors (MLT) in Crete, Greece, in terms of incidence and mortality rates, and identify the high-risk areas on the island (i.e., municipalities), to suggest public health measures. Data were obtained from the Cancer Registry Center (CRC) and included all cases of MLT for the period 1992–2013 in Crete. Age-standardized incidence rates (ASIR), age-specific incidence rates (ASpIR), age-standardized mortality rates (ASMR), and age-specific mortality rates (ASpMR) were estimated. For the study period (1992–2013), incidence and mortality showed an increasing trend. Mean ASIR was found 15.3/100,000/year and mean ASMR 8.6/100,000/year. Age groups 65–69 and 75–79 years among men presented the highest rates of (ASIR = 39/100,000/year) and among women age groups of 75–79 and 80–84 years a mean ASIR (22/100,000/year). The five-year survival rate of MLT was 50% and the ten-year survival rate was 47% for both genders. Risk factors that were identified included personal history of cancer, family history of MLT or other cancer, degree of relationship, smoking, and obesity. Some municipalities of Crete were found to be high-risk areas for MLT, while differences were detected in incidence and mortality rates, and annual rate change among them. Estimated variation indicates further increase probably due to the lifestyle of the residents, economic crisis, and inadequate preventive measures.

## 1. Introduction

Liver disease is now the second leading cause of years of working life lost in Europe, after ischemic heart disease, with malignant liver tumors classified as the second most common cause of death in developing countries and sixth in developed countries [1]. In 2020, primary liver cancer was the sixth most common tumor in terms of incidence and the third most lethal tumor in terms of mortality [2]. Hepatocellular carcinoma accounts for more than 80% of all primary liver cancers. Although rarer than hepatocellular carcinoma, cholangiocarcinoma, arising from the bile duct epithelium, confers an even poorer prognosis due to late diagnosis [3,4]. Only 20% of patients with cholangiocarcinoma are eligible for surgical resection, with a 5-year survival rate of less than 10% for all patients.

In Greece, the incidence was estimated at approximately 8.3/100,000/year for men and 2.9/100,000/year for women. According to the results from a study in Crete, the incidence rate for men was found to be 24/100,000/year and 10/100,000/year for women [5,6]. However, we need more recent data in Greece.

The need to focus on liver tumors is supported by the rapid changes in the prevalence of Non-Alcoholic Fatty Liver (NAFLD) and Non-Alcoholic Steatohepatitis (NASH), a condition that it is strongly associated with metabolic syndrome, obesity, and diabetes mellitus. According to a Cretan study, the incidence of cirrhosis has remained constant over the years, but the incidence of hepatocellular carcinoma increased during the last decade. This paper reports that risk factors for cirrhosis and hepatocellular carcinoma have changed over the past 25 years in Crete and that NAFLD is continually increasing and is a prominent risk factor for cirrhosis and hepatocellular carcinoma [7]. Of course, direct comparisons between the risk factors and the liver cancer rates on 1990s and 2000s could not be attempted, due to the fact that these data are not considered comparable. Only temporal trends can be observed and discussed.

In Crete, Greece, an increased trend of metabolic syndrome and obesity has been observed by the Regional Cancer Registry (CRCRC). Thus, we decided to investigate further the real burden of malignant liver tumors and the present study attempted to analyze available data for malignant liver tumors in Crete, to assess the morbidity burden and to explore possible risk factors for the occurrence of the disease among Crete’s municipalities. Based on the empirical observation of clinical data, we assumed that an increase in the malignant liver tumor indicator rates and possibly geographical variations in the island of Crete would be observed. The ultimate goal of this study was to suggest evidence-based public health measures to improve the health status of the population.

## 2. Methods

### 2.1. Setting and Study Population

The current study was conducted by the CRC team, utilizing data on new cases and deaths from malignant liver tumors during the period 1992–2013 in Crete. CRC is a member of the International Association of Cancer Registries (IACR) and the European Network of Cancer Registries (ENCR). Its direction is to systematically and thoroughly record incidence and deaths of patients with malignant neoplasms on the island in order to propose a valid and accessible prevention and management methodology. It was founded in 1992 in Crete and covers all the counties of the island, with a population estimated about 623,000 inhabitants. In 2014, with the support of the Region of Crete, a new methodology and action plan was introduced through the creation of an upgraded digital Cancer Monitoring (-recording) System (CMS) based on the CanReg5. The new CRC digital system (CMS) is suitable for the admission and management of “big data”, in accordance with international standards for disease coding (ICD10) and personal data protection. After its implementation, we digitalized all data from 1992 to 2014, till today.

### 2.2. Criteria for Inclusion in the Study

In the present study, 3130 new cases of malignant liver tumor were processed, attending the following inclusion criteria: (1) cases of primary malignant liver tumor, (2) permanent residence of Crete, for at least one decade, and (3) cases with histologic or cytologically documented diagnosis of malignant liver tumor. Cases with no information on the demographic and medical profile of patients were excluded from the present study, as well as cases with no information on the variables included in the regression model. The final number of patients included in this study was 1942 patients who met all the aforementioned criteria, while only a 4.2% of the cases did not satisfy the inclusion criteria (See Figure 1).

### 2.3. Variables Recorded in the Study

The variables used were: morphological diagnosis, age at diagnosis (number and age groups: 15–39, 40–49, 50–59, 60–69, 70–79, 80–99 years), stage at diagnosis (I, II, III, IV, Unknown), individual’s medical history of cancer (No, Yes, other than malignant liver tumor, Unknown), family medical history (No, Malignant liver tumor, Other non-malignant liver tumor, Unknown), affinity degree for patients with a family history of CNS liver (1st, 2nd, Unknown), smoking (Non-smoker, Smoker, Unknown), alcohol (Non-consumption, Consumption, Unknown), obesity–metabolic syndrome (Yes, No, Unknown). The category “unknown” in all variables refers to either not finding this information in the medical record or in cases where the biopsy did not specify the documentation of the information.

### 2.4. Measures

To assess the disease burden, the following incidence and mortality rates were calculated for the whole of Crete, per municipality:Age-standardized incidence rates (ASIR)Age-standardized mortality rates (ASMR)

The standardized incidence and mortality rates were calculated by the direct standardization method, using the 2001 European population as the model population (mean census year 1992–2013). The indices were calculated for the period 1992–2013 as a whole but also per year based on the following formula:

∑(crude rate for age group × standard population for age group)/∑standard population

### 2.5. Statistical Analysis

The analysis was performed using two pieces of software (Stata, ArcGIS, Esri, Redlands, CA, USA), at a statistical significance level α = 0.05. In addition to the partition tests and the descriptive statistics and graphs, the standardized mortality and impact indicators per municipality of Crete were calculated. Prediction and spatiotemporal analysis through mathematical polynomials and interpolation models were also applied. Specifically, the following tests were performed: spatial mean/median, spatial eclipse, spatiotemporal projection models, hot spots analysis, and kriging interpolation Bayesian models based on literature [8].

### 2.6. Ethical Approval

The CRC holds a license from the Hellenic Data Protection Authority (protocol number: 960/11-8-2009) and the University of Crete Committee of Ethic in Research and has adopted the rules for collecting, managing, and processing sensitive and personal data. All information was recorded using a cryptographic coding system in accordance with federal law principles and stored in the CRC server. No personal or individual-level data will be published.

## 3. Results

### 3.1. Morbidity and Mortality from Malignant Liver Tumor

According to the CRC’s data, during the period of 1992–2013, malignant liver tumors are listed in fifth position in terms of neoplasm incidence in Crete, with 15.3 new cases/100,000/year, and sixth in both sexes with a mortality rate of 8.65/100,000/year, with a higher rate in men. Incidence starts to increase significantly by the age of 45 years. In males, the most vulnerable age group is 65–69 years, with an ASIR of 38/100,000/year, and 75–79 years with an ASIR of 39/100,000/year. In females, respectively, the most vulnerable age group is 75–79 and 80–84, with an ASIR of 22/100,000/year. Detailed demographic and clinical characteristics of our sample are depicted in Table 1.

Figure 2A demonstrates the gradual increase in incidence rates in both sexes and per sex, starting from an ASIR of 20/100,000/year for men in 1992 and reaching an ASIR of 24/100,000/year in 2013. Correspondingly, for women the incidence is 5/100,000/year in 1992 and reaches 10/100,000/year in 2013. In both genders, the ASIR is 12/100,000/year in 1992 and 17/100,000/year in 2013. Furthermore, Figure 2B shows an obvious and gradual increase in mortality from malignant liver tumor in both sexes. In 1992, the age-standardized mortality rate is recorded at 9/100,000/year for men, 2/100,000/year for women, and 5/100,000/year for men and women. In comparison, 2013 saw an increase for both genders, with 12/100,000/year for men, 6/100,000/year for women, and 10/100,000/year overall (men and women,) with expected growth trends for the coming years.

### 3.2. Types of Malignant Liver Tumors

Another parameter that was studied is the frequency based on morphological diagnosis (Figure 3). Specifically, for the different types of malignant liver tumor, the incidence rate, taking into account the histological type, is the following: LCC (liver cell carcinoma): 61.1% males, 31% females; intrahepatic bile duct carcinoma: 26.4% males, 53.9% females; hepatoblastoma: 0.4% males, 0.3% females; angiosarcoma: 0.3% males, 0.2% females; other sarcomas of liver: 0.2% males, 0.4% females; other liver carcinomas: 11.6% males, 14.2% females. This distribution presented no statistical differences within the island municipalities, nor over time (*p*-value < 0.05).

### 3.3. Therapy

Further findings include the type of therapy. Radiotherapy was performed in 4.8% of stage I and II patients, in 2.8% of stage III patients, in 8.7% of stage IV patients, and 2.8% in patients of unknown stage (X). Respectively, chemotherapy was experienced by 22% of stage I patients, 31% of stage II patients, 32% of stage III patients, 31% of stage IV patients, and 20% of stage X patients. Tumor removal surgery was experienced by 70% of stage I patients, 55% of stage II patients, 29% of stage III patients, 8% of stage IV patients, and 18% of unknown stage X patients.

### 3.4. Survival Rate

Another observation derived from the present study focuses on the issue of survival according to the year of diagnosis of malignant liver tumor. During the first year of diagnosis, there was no difference in survival between the two sexes (68% survival for men, 68% for women, and 68% for men and women overall). Over five years, survival for men declined to 54%, for women to 48%, and for men and women to 50% overall. Over a decade, survival for men declined to 47% and for women to 44%. At 15 years, survival declined for men to 42%, for women to 39%, and for men and women to 41% overall. Furthermore, in the age group 60–69 years, the survival rate in men decreased to 13.9%, and in women 12.5%, in the age group 70–79 years survival decreased further in both sexes (men 8.2%, women 6.6%), and in the last age group of the study, 80–99 years, the survival rate in men decreased to 5.7% and in women to 1.8%.

### 3.5. Geographical Distribution

In the present study, the ASIR (age-standardized incidence rate) and the ASMR (age-standardized mortality rate) were assessed in the counties and municipalities of Crete (Figure 4A). As for ASIR in the counties of Crete, the municipalities with the highest impact were the municipalities of Heraklion, Rethymnon, Chania, and Ierapetra with ASIR 18–19/100,000/year. In the same figure, the ASMR mortality index of counties and municipalities is illustrated, which was calculated using the standardization method in order to smooth out [PK8] the differences in population structure. The kappa statistic homogeneity spatial index was used, which is equal to 0.45, and thus showed significant differences in the geographical distribution of mortality indices. It can be noticed that there is a high incidence of deaths due to malignant liver tumor in the municipalities of Heraklion and Rethymnon with ASMR of 9–9.5/100,000/year.

In Figure 4B, the average annual mortality as shown by the spatiotemporal analysis of data for the years 1992–2013 is distributed. The municipalities of Heraklion and Rethymnon, represented in red, appear to have the highest death rates per year. The lowest average annual mortality rates occur in 11 municipalities. The spatiotemporal trend of the distribution seems to have a clockwise development, and shifts annually in different regions. More specifically, the spatiotemporal analysis of ASMR per year shows that during the first years there were lower rates in the counties of Heraklion and Rethymnon. On the contrary, in 2013 the counties of Heraklion and Rethymnon appear to have the highest rates, followed by the counties of Chania, Kissamos, and Ag. Vasileios. No particular change is observed in the trends up until 2017.

### 3.6. Relative Risk (RR)

Table 2 lists the core risk factors of low survival of patients with malignant liver tumor. Specifically, patients with medical history of cancer (apart from malignant liver tumor) present 2.4-times higher risk of low survival (95%CI 2.2–2.6), while those with a history of hepatitis B and C have an RR of 2.8 (2.1–3.5). Similarly, higher risk is present in those with a family medical history of malignant liver tumor (RR 2.3, 2.0–2.6), or other type of cancer (1.8, 1.4–2.2), especially first-degree relatives. A high RR is also presented for obese patients (1.7, 1.1–2.3). Lastly, patients who consumed alcohol presented 3.1-times higher risk (2.4–3.8).

Figure 5 depicts the relative risk of low survival of patients with malignant liver tumor in the island, from 1992 to 2013. As shown, the areas with high relative risk (RR) are Heraklion, Hersonissos, Archanes, Asterousia, Rethymnon, Chania, Kissamos, Ierapetra, and Sitia with an RR = 2.8–3.1. These are 8 out of 23 municipalities. The municipalities with average relative risk are the municipalities of Anogeia, Mylopotamos, Malevizi, Gortyna, Phaistos, Minoa-Pediados, Viannos, Apokoronou, Sfakion, Platania, Kantanou, Selinou, and Ag. Nicholaos (n = 13). Two municipalities remain, with low relative risk, the municipalities of Lasithi Plateau and Ag. Vasileios.

## 4. Discussion

### 4.1. Main Findings

This study revealed an upward trend of malignant liver tumors in Crete over the period 1992–2013, and this was visible across all of this island. However, four municipalities (Heraklion, Rethymnon, Chania, and Ierapetra) were found to be more affected. The study also added to the discussion of several potential determinants of malignant liver tumors including the history of hepatitis B and C, history of coexisting cancer, family history, alcohol consumption, and fatty liver.

### 4.2. Reading Local Data with the Existing Literature

In Greece, a limited number of epidemiological studies have been carried out. Hadziyannis et al. [9] report the significant association of HBV and HCV in the occurrence of malignant liver tumors in Greece. Kamposioras et al. carried out another important cancer screening study in Greece in 2008 with the purpose of estimating the incidence of cancer in the Greek population and the difficulties for early diagnosis [10]. Crete constitutes a homogeneous population geographical area with no significant differences in population composition from the rest of the country, so it was not expected that there would be a difference compared to the rest of Greece in terms of incidence of viral hepatitis as well as of malignant liver tumors, especially in particular areas of the island [8]. These results strengthen the hypothesis that possible factors related to the habits of the residents and the environment may be the cause of the increased incidence of the disease. Cancer of the gastrointestinal tract in its whole should be approached with bio-ethic, geographical, and lifestyle co-determinants that can act in interaction or in synergy [11].

Our study has shown that the incidence of malignant liver tumors in Crete has increased from 1992 to 2013. In Sung et al.’s [2] study, liver cancer constituted 8.3% of all cancers in 2020. Liver cancer covers the fifth position in terms of global incidence and second in terms of mortality for men. It is the most common cancer in eleven geographically diverse countries, in Eastern Asia (Mongolia, which has rates far exceeding any other country), South-Eastern Asia (e.g., Thailand, Cambodia, and Viet Nam), and Northern and Western Africa (e.g., Egypt and Niger).

Regarding the histological types in the literature, HCC accounts for 70–80% internationally, followed by ICC intrahepatic bile duct carcinoma for 15%, with the remaining 5% being other hepatic neoplasms [12]. However, the results show that in Crete HCC has lower rates, about 60%, while ICC is increased, mainly in women, up to 53%, which are similar to rates in other countries such as France, Italy, and, Thailand, differing from the rest of Greece. In the study of Karageorgos et al. [7], the incidence of cirrhosis has not changed over the years, but the incidence of HCC has increased during the last decade and the same holds for its risk factors. The initial high hepatitis C virus association has significantly decreased, with alcohol now ranking first among risk factors. Additionally, non-alcoholic fatty liver disease is rapidly increasing as a prominent risk factor for cirrhosis and HCC in Crete. Similar findings were also discussed in Markakis et al. (2022) [13].

Focusing on the determinants of liver malignant neoplasms, this study revealed that history of hepatitis HBV/HCV, history of cancers other than malignant liver tumor, and family history of malignant liver tumor or other type of cancer. Specifically, family history from a first degree family member, first degree of alcohol consumption, and obesity [12,14,15,16,17,18,19,20] are widely reported as risk factors. In these articles, there is a discussion of the development of malignant liver tumor due to obesity, without preceding cirrhosis, simply the so-called fatty liver, the result of metabolic syndrome, which is very common nowadays. The mechanisms by which hepatocellular carcinoma develops are not yet known and are yet to be investigated. The changes in lifestyle of the Cretan population and the increase incidence rates of obesity, diabetes mellitus, and metabolic syndrome during the past few decades may have an impact on the findings of this study [21]. These changes in the Cretan lifestyle [21] could also explain the geographical differences observed in the incidence distribution, which was found to be at higher levels in the urban or semi-rural regions. These variations should not be fully associated with the population and cases of low distribution in some regions, since we have applied age-standardization that has smoothed these differences. The authors believe that further studies should be performed in order to explore the causal factors and determinants of this outcome. However, special focus should be given to lifestyle and related risk factors, including obesity [21].

### 4.3. Strengths and Limitations

The present study establishes the first complete study on morbidity and mortality from malignant liver tumor in Crete, which is supported by population data of two decades (1992–2013). The size of the sample and the adoption of European and international standards for the collection, classification, and analysis of cancer data provide particular value and external credibility in this study. The findings of the study can provide essential guidance for the planning and organizing public health programs, prevention, and diagnostic measures for malignant liver tumor in the Crete area.

Nevertheless, this study and its findings need to be discussed under some limitations. The findings are directly expressing the current burden in the Region of Crete and cannot be spontaneously generalized at a national level. Another limitation is the full coverage of the recording of all malignant liver tumor, which relates to the likelihood of ‘missing’ incidents. This is a possibility that is present in all epidemiological studies. Additionally, it is likely that the increase in tendency over the course of the twenty years is due to the continuously decreasing trends of patients with malignant liver tumor that are leaving Crete to seek diagnosis and treatment elsewhere. This limitation is however considered to be small in this study, due to the open population cohort we managed to establish. Furthermore, the likelihood of information error when transferring data from hospital and registry files should also be noted. This error percentage is not expected to be high (<5%) as all quality indexes meet the IARC criteria of 98%. In order to limit the weaknesses and potential errors of the study, weighted morbidity and mortality indicators were used at all stages of the analysis (thus equating any population differences), after complete qualitative control and classification of the data was carried out (ensuring the high quality of the available data of 98%). This percentage is estimated for the dataset used for the current analysis, while the 2% remaining refers to missing data in this dataset. These missing data referred to lack of information on body mass index and body surface area (BMI, BSA), which were not used in the reported results. Lastly, it should be stressed that direct comparisons between the rates on 1990s and 2000s were not attempted, since these data are not comparable due to variations in data collection, means of diagnosis, etc.

### 4.4. Impact of the Study

According to the findings, it seems that in Crete, malignant liver tumors show high incidence and mortality rates, so it is important to continue this observational study and to utilize the present results at island level in terms of health planning (patients, professionals, and health care providers). A mandatory step and priority for our work will be to continue the registration of malignant liver tumors for the coming years, as well as to investigate the risk factors, which may be responsible for the inequalities between municipalities.

The results of this study provided information on the burden and geographical distribution of cancer. Based on this information, Public Health initiatives can be developed to prevent and control cancer. At the same time, these data are a reliable way to evaluate existing pre-symptom screening programs, the effectiveness of treatments, the survival time of oncology patients, and the quality of treatment and care they receive. Finally, it is important to emphasize that proper management of the problem will bring significant financial relief to health costs, as the cost of care for oncology patients is one of the highest in the health sector. In most European centers, hepatocellular carcinoma surveillance falls under the responsibility of secondary care. We adapt the strong recommendation raised by the recent report of The Lancet/EASL commission which stressed that “A fundamental shift must occur, in which health promotion, prevention, proactive case-finding, early identification of progressive liver fibrosis, and early treatment of liver diseases replace the current emphasis on the management of end-stage liver disease complications”. If we consider that “COVID-19, alongside imposing delays in diagnostic pathways of liver diseases, has brought overlapping metabolic risk factors and social inequities into the spotlight as crucial barriers to liver health for the next generation of Europeans”, it is evident that there is an urgent need for targeted preventive and management measures for liver cancer and liver disease in general [22].

## 5. Conclusions

The increased incidence of and mortality from malignant liver tumors in Crete, as well as the difference in impact and mortality in Crete compared to the rest of Greece (similarity to Italy, Spain) were reported in this study. Taking into consideration various points mentioned in the study, an increase in the incidence and mortality trend is expected in the future, while the increased incidence of ICC in Cretan women is worth mentioning.

These data provided information on the burden and geographical distribution of the cancer. Decisions and measures need to be informed to contribute to reduce mortality. National cancer registries informed from local data pools need to start tracing all cancer cases to inform services and enhance health planning.

## Figures and Tables

**Figure 1 ijerph-19-10166-f001:**
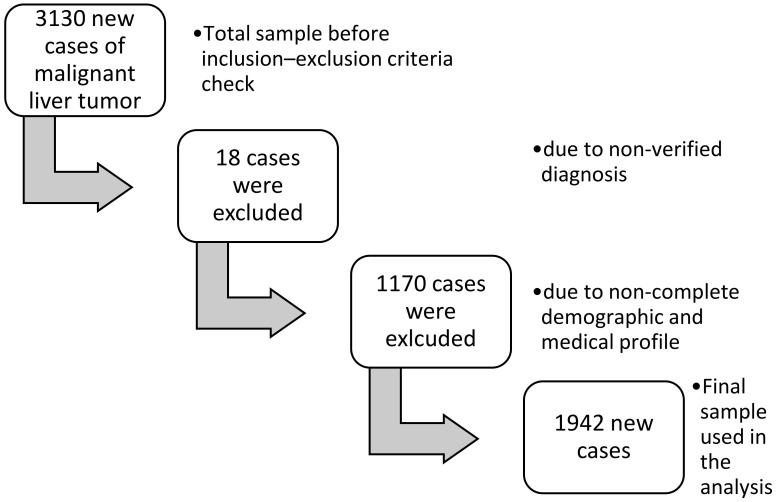
Flow chart depicting the sampling based on inclusion and exclusion criteria.

**Figure 2 ijerph-19-10166-f002:**
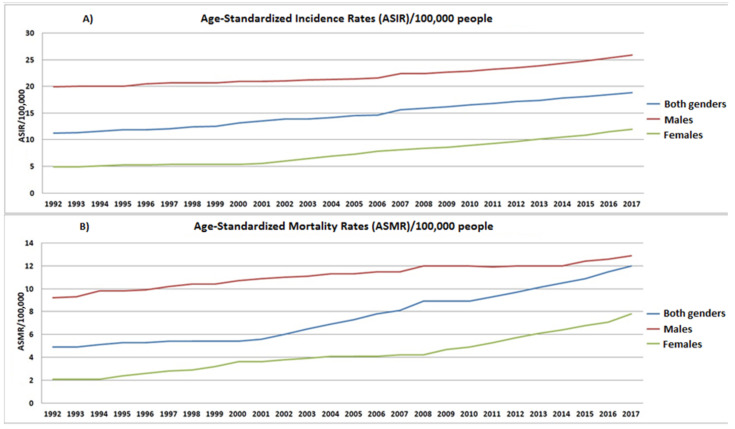
Gender-based distribution of Age-Standardized Incidence (**A**) and Mortality (**B**) Rates per 100,000 people/year for the time period of 1992–2017.

**Figure 3 ijerph-19-10166-f003:**
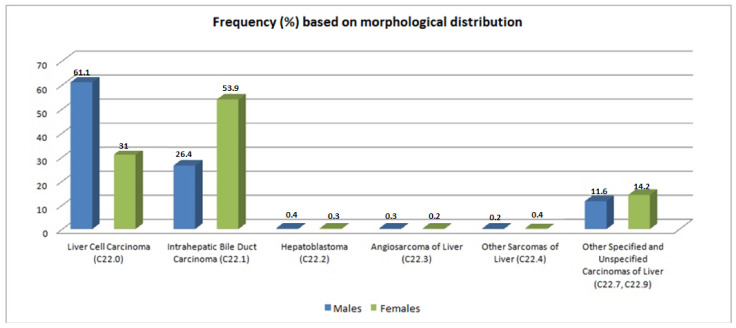
Frequency (%) of cases based on morphological distribution.

**Figure 4 ijerph-19-10166-f004:**
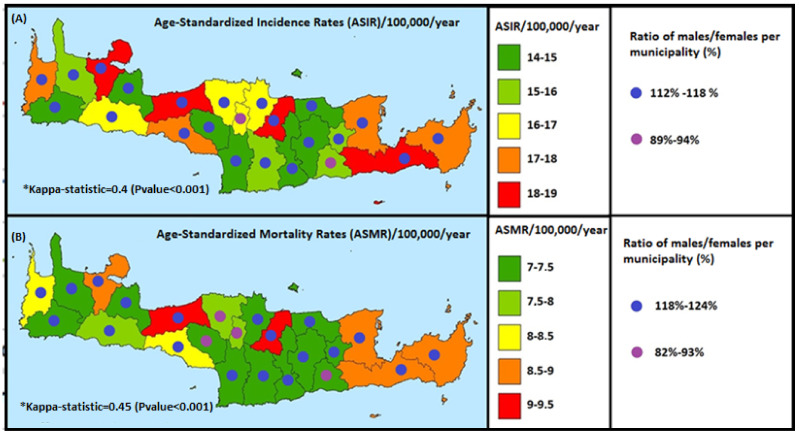
Gender-based distribution of Age-Standardized Incidence and Mortality Rates per 100,000 people/year for the time period of 1992–2017.

**Figure 5 ijerph-19-10166-f005:**
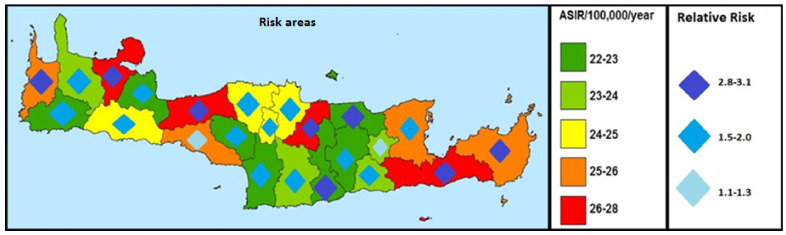
Risk areas of low survival of patients with malignant liver tumor according to the identified risk factors. (All RRs are statistically significant, *p*-value < 0.05).

**Table 1 ijerph-19-10166-t001:** Demographic and medical profile of patients with malignant liver tumors in Crete, for the time period of 1992–2013 (n = 2022).

Characteristics	With Malignant Liver Tumor		*p*-Value
	n = 2022	%	
**Sex**			0.03
*Male*	1490	73.7	
*Female*	532	26.3	
**Age at diagnosis**	64	3.4	<0.001
**Stage at diagnosis**			0.04
*I*	495	24.5	
*II*	907	44.8	
*III*	396	19.6	
*IV*	126	6.2	
*Unknown*	98	4.8	
**Obesity (based on BMI)**	495	24.5	0.04
**Individual medical history of cancer**			0.03
*No*	719	35.5	
*Yes (apart from malignant liver tumor)*	1280	63.3	
*Unknown*	23	1.1	
**Individual history of Hepatitis B and C**	632	34.2	0.04
*Family medical history*			0.02
*No*	211	10.4	
*Malignant liver tumor*	787	38.9	
*Other malignant tumor (apart from malignant liver tumor)*	979	48.4	
*Unknown*	45	2.2	
**Degree of relationship (family history of malignant liver tumor, n = 787)**			<0.001
*1st*	557	70.7	
*2nd*	208	26.4	
*Unknown*	22	2.8	
**Smoking**			0.01
*Non-smokers*	582	28.7	
*Smokers*	1338	66.1	
*Unknown*	102	5.0	
**Alcohol**			
*Non-consumption*	671	33.2	0.01
*Consumption*	1275	63.1	
*Unknown*	76	3.7	
**Routes of diagnosis**			0.03
*Emergency clinic*	992	49.1	
*Other hospital clinics*	113	5.6	
*Referral from a general practitioner*	516	25.5	
*Another outpatient clinic/diagnostic*	208	10.3	
*Unknown*	193	9.5	

**Table 2 ijerph-19-10166-t002:** Relative risk of low survival of patients with malignant liver tumor and association with risk factors.

Characteristics	Relative Risk (95%CI)	*p*-Value
**Individual medical history of cancer**		<0.001
*No*	1	
*Yes (apart from malignant liver tumor)*	2.4 (2.2–2.6)	
*Individual history of Hepatitis B and C*	2.8 (2.1–3.5)	0.03
**Family medical history**		0.02
*No*	1	
*Malignant liver tumor*	2.3 (2.0–2.6)	
*Other type of cancer (apart from malignant liver tumor)*	1.8 (1.4–2.2)	
**Degree of relationship**		<0.001
*2nd*	1	
*1st*	2.9 (2.8–3.1)	
**Obesity (based on BMI)**		0.04
*No*	1	
*Yes*	1.7 (1.1–2.3)	
**Alcohol consumption**		0.01
*Non-consumption*	1	
*Consumption*	3.1 (2.4–3.8)	0.03

## Data Availability

All data generated or analyzed during this study are included in this published article. No raw data are available due to the private data regulation, but further results could be shared for research purposes, upon request.

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
