# Peer review of "Reporting Liver Cancer Trends in the Island of Crete, Greece: Results from a Geo-Epidemiological Study"

_ijerph, 2022, doi:10.3390/ijerph191610166_

Round 1

Reviewer 1 Report

-The work aimed to assess the “burden” of Malignant liver tumors (MLT) in Crete, Greece  in terms of incidence and mortality rates and identify the high-risk island county areas to suggest public health measures. The study is interesting and important, showing valuable data. My concerns are:
-The abstract should be improved. Please give a brief background before giving your aims. In addition, the phrase "Further studies are needed to form awareness measures to reduce the disease “burden” is more adequate in the discussion than in the abstract.
-Figure 1 legend was inserted under Table 1(line 138). Then, figure 1 is placed with a legend. Please revise well all tables and figures and clearly describe them in the text. The way it is shown is not that clear. 
-Moderate English changes are required

Author Response

-The work aimed to assess the “burden” of Malignant liver tumors (MLT) in Crete, Greece  in terms of incidence and mortality rates and identify the high-risk island county areas to suggest public health measures. The study is interesting and important, showing valuable data. My concerns are:

-The abstract should be improved. Please give a brief background before giving your aims. In addition, the phrase "Further studies are needed to form awareness measures to reduce the disease “burden” is more adequate in the discussion than in the abstract.

Re: Thank you very much for all your comments. We’ve revised the abstract according to your remark.

-Figure 1 legend was inserted under Table 1(line 138). Then, figure 1 is placed with a legend. Please revise well all tables and figures and clearly describe them in the text. The way it is shown is not that clear.

Re: Thanks once again. We’ve relocated figures’ titles above each figure in order to follow the same format with the tables.

-Moderate English changes are required

Re: Entire manuscript was checked and revised by a native English speaker. Thank you for the recommendation.

Reviewer 2 Report

This is an interesting epidemiological study on liver cancer in Crete. The spatial and temporal analyses are particularly interesting. I only have a few minor issues and suggest to revise the English to ensure readability and avoid confusion, particularly in the discussion. 

Minor points:

In introduction:

The reference for the impact of liver cancer is wrong, the numbers you cite come from Global Cancer Statistics published on CA and/or the global burden of disease, please cite the most recent and apropriate. 

Comparison between data from a 1990s study in Crete (rates seem high is the standardization comparable) and 2010 in Greece is misleading, it should be clearly stated that these data are not comparable.   

Methods and results

Line 72 state when the new method of data collection was introduced and since when it influences data. 

In the criteria you state that you start from ~3000 liver cancer cases of which ~2000 satisfy criteria, and then state only 4.2% didn't satisfy criteria. Please specify (possibly table or schematic) selection/exclusion process.

In the results section relative to figure 2 and the cancer subtypes, the relative distribution of cancer subtypes is given. IT would be usefull to know whether this distribution is homogeneous over the whole period under study or whether it evolved over time and if there are differences is the major geographic areas subdivisions, since the different subtypes have different aetiologies and survival outlooks. 

Discussion

In the spatial analysis, the more densely populated/urbanised areas seem to have greater incidence mortality, could this be an effect of low numbers random variation (in low density areas), or would the authors consider this more lifestyle behavioral? I would like this to be addressed further in the discussion.

In the limitation paragraph I would like to read details on data completeness and percentages of items set to unknown in the overall data if possible.

Author Response

This is an interesting epidemiological study on liver cancer in Crete. The spatial and temporal analyses are particularly interesting. I only have a few minor issues and suggest to revise the English to ensure readability and avoid confusion, particularly in the discussion.

Re: Entire manuscript was checked and revised by a native English speaker. Thank you for the recommendation. We’ve also provided responses to the minor points, below.

Minor points:

In introduction:

The reference for the impact of liver cancer is wrong, the numbers you cite come from Global Cancer Statistics published on CA and/or the global burden of disease, please cite the most recent and apropriate.

Re: Thank you very much for this remark. This is the latest official report we could find (numbers are reported for 2020). We’ve checked the numbers and revised the paragraph accordingly (we omitted one sentence and checked the other which were correct).

Comparison between data from a 1990s study in Crete (rates seem high is the standardization comparable) and 2010 in Greece is misleading, it should be clearly stated that these data are not comparable.  

Re: Thank you for the comment. We have now added a clear statement in the introduction, as you commented. Specifically, we stated that “Of course, direct comparisons between the risk factors and the liver cancer rates on 1990s and 2000s could not be attempted, due to the fact that these data are not considered com-parable. Only temporal trends can be observed and discussed.” (Lines 56-59)

In addition to that, we also added a similar statement in the study limitations (Lines 363-367). We mentioned that “Lastly, it should be stressed that direct comparisons between the rates on 1990s and 2000s were not attempted, since these data are not comparable due to variations in data collec-tion, means of diagnosis, etc.”

Methods and results

Line 72 state when the new method of data collection was introduced and since when it influences data.

Re: After the new monitoring system implementation, we digitalized all data from 1992 to 2014, till today. There is no impact, or influence in the data presented in this paper, since all these data were collected and digitalized in the same way. We’ve made minor revisions in Lines 78-84.

In the criteria you state that you start from ~3000 liver cancer cases of which ~2000 satisfy criteria, and then state only 4.2% didn't satisfy criteria. Please specify (possibly table or schematic) selection/exclusion process.

Re: We have now explained it and added a new figure (Figure 1), which describes all the steps followed for the sampling.

In the results section relative to figure 2 and the cancer subtypes, the relative distribution of cancer subtypes is given. IT would be usefull to know whether this distribution is homogeneous over the whole period under study or whether it evolved over time and if there are differences is the major geographic areas subdivisions, since the different subtypes have different aetiologies and survival outlooks.

Re: Thank you for the valuable remark. We had checked for differences in this distribution and found no significant variations among the geographical areas (municipalities) or time periods. Only slight differences were observed but with no statistical or clinical significance. Therefore, we added the following sentence. “This distribution presented no statistical differences within the island municipalities, nor over time (pvalue<0.05).” (Lines 172-173)

Discussion

In the spatial analysis, the more densely populated/urbanised areas seem to have greater incidence mortality, could this be an effect of low numbers random variation (in low density areas), or would the authors consider this more lifestyle behavioral? I would like this to be addressed further in the discussion.

Re: These changes in the Cretan lifestyle could also explain the geographical differences observed in the incidence distribution which was found to be at higher levels in the urban or semi-rural regions. These variations should not be fully associated with the population and cases low distribution in some regions, since we have applied age-standardization that has smoothed these differences. The authors believe that further studies should be performed in order to explore the causal factors and determinants of this outcome. However, special focus should be given on lifestyle and related risk factors, including obesity.  We have now added this explanation in the discussion. Lines 330-337.

In the limitation paragraph I would like to read details on data completeness and percentages of items set to unknown in the overall data if possible.

Re: We have now added details on the completeness percentage. Thank you.

Specifically, we mentioned that: “This percentage is estimated for the dataset used for the current analysis, while the 2% remaining percentage referrers to missing data in this dataset. These missing data referred to lack of information on body mass index and body surface area (BMI, BSA), which were not used in the reported results.” [Lines 360-362)
